
# Comparing the evolution of ESA versus NASA technology transfer
# approach: market and public demand drivers
Giorgio Petroni[1], Benedetta Pini[2], Serena Filippelli[2], Alberto Petroni[2]
[1] INAF – Istituto Nazionale di Astrofisica, Milan, I20100, Italy
[2] Department of Engineering and Architecture, University of Parma, Parma, I43100, Italy
*Correspondence to*: Alberto Petroni (alberto.petroni@unipr.it)
**Abstract.** The growth of space activities has experienced rapid expansion in the last twenty years, largely driven by the
transfer of technology. This process has had not only economic and social effects but also important political and military
implications.
The primary entities responsible for sharing scientific and technological knowledge have been the prominent space
agencies. This article seeks to compare the approaches developed by NASA and ESA throughout the years. The
comparison reveals significant differences between the two agencies in terms of their goals and the methods they employ
to achieve them. These disparities can even be traced back to the legislation that established each respective agency.

**Keywords**: Space technology transfer, technology transfer process, market-driven technology
transfer, ESA infrastructure Galileo and Copernicus programs, technology network diffusion.

**1 Introduction**
In the context of the space sector, the history of technology transfer has its roots in the early days of space exploration.
Initially driven by government-funded programs and national space agencies, technology transfer in space began with the
exchange of knowledge and expertise between countries engaged in space missions.
It is noteworthy that even after NASA's establishment in February 1958 as a civilian organization, knowledge and
experience exchange with the military persisted, primarily focusing on launch systems. A notable example is the
development of the Saturn V launcher, utilized in the Apollo 11 mission and lunar landing in July 1969, which was guided
by Werner Von Braun, a consultant to the US Armed Forces following World War II (Spangenburg et al., 2003).
During the Space Race between the United States and the Soviet Union in the mid-20th century, significant advancements
were made in rocketry, satellite technology, and space exploration. The technologies developed for space missions, such
as propulsion systems, communication satellites, and remote sensing capabilities, had applications beyond the space
sector (Johnson, 2012; Logsdon, 2011).
Following the "Space Race," space cooperation commenced through bilateral agreements signed by the USA and USSR,
involving their respective allies or partners. These agreements facilitated collaborations between NASA and ESA, NASA
and NASDA (the former Japanese space agency), as well as NASA and the Indian Space Research Organisation (ISRO).
Subsequently, additional significant bilateral agreements on space cooperation emerged, involving the Russian Space
Agency "Roscosmos," the Chinese National Space Administration (CNSA), and the South Korean Aerospace Research
Institute (KARI) in the mid-1970s. This trend continued with the participation of large African countries like Egypt and

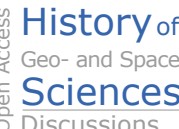

South Africa, as well as Latin American countries such as Argentina, Brazil, and Mexico. Consequently, a substantial
knowledge transfer process in the space field ensued.
As space exploration expanded, governments and space agencies recognized the potential for commercializing space
technologies and transferring them to other industries. The space sector became a source of innovation and a catalyst for
technological advancements in fields such as telecommunications, Earth observation, materials science, and navigation
systems (Schmitt, 2004).
Government agencies like NASA in the United States and ESA in Europe played a pivotal role in technology transfer by
promoting collaboration between the space industry, research institutions, and private companies. They established
partnerships, licensing agreements, and cooperative programs to facilitate the transfer of space technologies for
commercial use and societal benefit.
Additionally, advancements in satellite-based Earth observation systems have led to the development of applications in
agriculture, urban planning, environmental monitoring, and disaster management. Data and imagery acquired from
satellites have been made accessible to various stakeholders, including governments, researchers, and businesses,
enabling them to make informed decisions and address pressing challenges.
In recent years, with the emergence of private space companies and the commercialization of space activities, technology
transfer in the space sector has taken on a new dimension. Companies like SpaceX, Blue Origin, and numerous startups
are pushing the boundaries of space technology and exploring innovative applications beyond traditional space missions.
The transfer of reusable rocket technologies, satellite miniaturization, and launch services to the private sector has fueled
the growth of the space industry and opened up new opportunities for collaboration and innovation.
Moreover, the flow of scientific and technical knowledge driving space activities has not solely originated from civilian
research laboratories (such as universities and academies) or military institutions, as witnessed during the early days of
the "Space Age." Many industrial enterprises and their suppliers, responsible for manufacturing and commercializing
space tools, have contributed significantly to the development of technical know-how.
Furthermore, the development of downstream space activities and satellite services has facilitated the widespread
dissemination of space-related technical knowledge to many developing countries. This ongoing process has led to a
proliferation of space agencies in the past 15 years, often promoting the establishment of research and training facilities
across various technological sectors of the industry.
Overall, the history of technology transfer in the space sector reflects the evolution of space exploration, the recognition
of space technologies' commercial potential, and the collaborative efforts between governments, space agencies, research
institutions, and private companies to transfer and utilize space-related innovations for economic, scientific, and societal
advancements.
These introductory remarks aim to capture the complexity of technology transfer as a catalyst for the development of
space activities. This article aims to elucidate the diverse approaches adopted by ESA and NASA in guiding technology
transfer and the significant ramifications it has had on political, military, and economic aspects (Williamson, 2014).

## 2 Actors and Channels of Technology Transfer in space

The generation and dissemination of space technologies have involved numerous actors, each contributing in their unique
ways. Firstly, astrophysicists have built a solid foundation of scientific knowledge pertaining to the structure of outer
space and the fundamental physical phenomena characterizing it, such as radiation, electromagnetic fields, gravity effects,



and energy source intensities (Rosenberg et al., 2014). This knowledge has played a vital role in enhancing the safety of
exploratory missions, including those involving human presence. Additionally, it has significantly advanced imaging and
detection technologies employed in Earth observations, exemplified by the utilization of lasers.
Furthermore, the military has made substantial contributions to "modern civil space" activities since their inception. A
notable example is the civilian application of the Global Positioning System (GPS) for mass consumer purposes, such as
in smartphones. Regarding launch systems, similar to the military's involvement in NASA's early launches mentioned
earlier, the European Space Agency (ESA) has benefited from the utilization of highly reliable launchers like Soyuz and
Vega, which trace their origins back to the original designs led by the Ukrainian engineer Korolev, a former colonel of
the Soviet Red Army.
Currently, there is a robust collaboration between NASA and the research laboratories of the US Navy for the
experimental construction of a "Solar Space Farm" aimed at generating electricity from solar panels deployed in outer
space. This partnership adds to the established cooperation between space agencies and armed forces of many nations in
the construction and operation of "dual-use" satellites.
It is crucial to acknowledge the significant contributions made by universities engaged in space research and education,
including those in developing countries. These institutions play a vital role in advancing space-related knowledge and
technologies.
Lastly, suppliers operating in various segments of the space system, including infrastructure (such as ground segments),
launch platforms, satellites, and technical components within the payload, serve as an important driving force for the
development and subsequent transfer of space technologies. Their expertise and innovation contribute to the continual
progress of the field.

## 2.1 Technological dynamics associated with space activities

The technologies employed within a space system are diverse, resulting in a highly complex and interconnected system.
To provide clarity, let us review the key components of this extensive technological repertoire. These include propulsion
and launch control technologies, the telematics guidance system that directs satellites along their designated trajectories
and controls their orbital paths, electronic apparatuses that regulate and manage equipment within the payload (including
instruments governing power production, temperature, and pressure), robotic technology, instruments utilizing various
forms of radiation (such as X-rays, gamma rays, and infrared) for astrophysical exploration and terrestrial observations,
as well as telecommunications systems.
Furthermore, space activities often require the utilization of materials capable of withstanding high temperatures and
pressures. For instance, satellite buses and space probes are constructed using such materials. Many of these technologies
already exist within industrial production systems and undergo upgrading processes to adapt them for space applications.
A notable example is the production of solar panels, which employ a "triple junction" scheme to generate the necessary
energy for the functioning of payload equipment. More recently, traditional solar panels have been integrated into a
sandwich structure that serves a dual purpose of generating electricity through photovoltaic processes and transforming
it into microwaves. This innovation is seen in experimental systems designed for the production and transmission of
electric power from space to Earth (Space Solar Power).
Lastly, within the context of technological dynamics associated with space activities, it is important to acknowledge the
role of small businesses as vehicles for industrial innovation transfer. These companies, often collaborating as suppliers
to major contractors (typically larger companies) responsible for significant space contracts, have facilitated the transfer

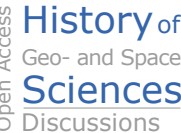

of technologies that have undergone further upgrading to non-space companies (Chebukhanova et al., 2022; Verbano et
al., 2012).

**2.2 "Technology Push" and "Demand Pull" Technology Transfer Processes**
In the last 15 years, there has been a notable acceleration in the widespread adoption of Earth Observation activities. This
surge has been primarily fuelled by the increasing demand for satellite services from public institutions and the rapid
growth of the consumer market for personal communication devices such as cell phones and iPads. This trend has had a
significant impact on various regions, including the United States, Europe, major Asia-Pacific countries, and, to a lesser
extent, developing nations (Guarnieri et al., 2016).
To meet the aforementioned demand, there has been a substantial rise in the production of even smaller satellites, which
have become increasingly commoditized, accompanied by a significant reduction in launch costs. As a result, there has
been a remarkable increase in the number of manufacturers, particularly small businesses, engaged in downstream space
activities, including satellite surveys, navigation, and communications (de Pippo et al., 2019).
However, it is crucial to avoid misconceptions regarding the assumed ease of technology transfer in space activities,
despite the significant advancements driven by "technology push" innovation. Operators in this sector, confronted with
the inherent complexity analyzed by B. Bozeman (2000) in each technology transfer program, face considerable
challenges even in Spin-in projects. These challenges arise from the necessity to design and construct technical tools
intended for operation in an environment vastly different from terrestrial conditions, characterized by microgravity,
radiation, high temperature, and pressure variations. Nevertheless, these constraints also lead to increased research efforts,
often resulting in paradoxical innovations that significantly enhance the effectiveness and efficiency of terrestrial
production activities. Noteworthy advancements have been made in sensor technology, the development of heat-resistant
materials, and the utilization of radiation in diagnostic healthcare equipment.
Empirical analysis further reveals the presence of demand-driven innovations, referred to as "demand pull innovations"
(Breton et al., 2019; Verbano et al., 2017). Examples include the growing utilization of satellite surveys to identify leaks
in territorial water networks supplying domestic consumption, the estimation of "equivalent water" to assess snow
accumulation in high mountains such as the Alps, monitoring subsidence on highways, and the satellite-based detection
of suitable areas in the seafloor for onshore wind turbine installation. These applications typically involve the upgrading
of existing technologies and are met with less resistance from producing companies and potential users. This is because
they facilitate the overcoming of socio-organizational barriers rooted in the maintenance of leadership, pre-existing
professional culture, and established work practices.

**3 NASA's Approach to Technology Transfer**
On July 29, 1958, President Eisenhower signed the establishing law that gave rise to NASA, the public agency responsible
for shaping not only American but also global space activities. This law outlined the objectives of the newly formed
organization, which succeeded the National Aeronautic Commission (NACA). It emphasized the primary mission of
generating scientific knowledge in the fields of aeronautics and space. Additionally, the law explicitly recognized the
potential for developing "scientific and engineering" applications based on this knowledge. The intention was to leverage

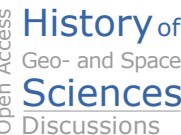

these applications for the benefit of the United States in collaboration with other nations, provided that they served
peaceful purposes (Anderson, 1981).
These provisions laid the foundation for NASA's future endeavors in technology transfer. It is worth noting that they were
enacted shortly after the Soviet Union's launch of Sputnik-1 on October 4, 1957, which caused widespread concern among
American citizens due to the perceived threat of a Soviet military attack.
The budgetary trajectory of NASA over time demonstrates an initial sharp incline leading up to the successful Apollo
Mission in 1969, followed by a subsequent, more gradual upward trend observed in the years 1993 and 1994. These
periods correspond to significant advancements in technology resulting from substantial investments in space research
and experimentation. The first period was characterized by the agency's intensive endeavors to achieve the remarkable
feat of landing on the moon, while the second period witnessed a renewed emphasis on manned astrophysical explorations,
garnering strong support from both President George W. Bush and President Bill Clinton. Many of these technological
breakthroughs have played a pivotal role in driving the process of technology transfer, extending beyond the confines of
the space sector. Notably, this process has continued to thrive even during periods of budgetary constraints for NASA in
subsequent years.
Official data provided by NASA indicates that as of the end of 2022, the agency had facilitated the creation of 2,100
spinoff companies and held a portfolio of 1,331 active patents.
Over time, it has become evident that the objectives laid out in the National Aeronautics Space Act have been largely
accomplished. The achievements are rooted in consistent choices made by the US federal government, facilitated by
NASA, which include:
a) Acknowledging that NASA's initial involvement in the production of innovative space applications should transition
to a market-driven approach, where the market becomes the primary force behind the development of such applications.
b) NASA's active efforts to promote the generation of scientific knowledge and the development of potential technical
applications through substantial investments in research and development.
c) The US government's commitment to fulfilling its responsibility of supporting the demand for space products and
services through industrial policy measures, aligning with its role.
To ensure effective implementation of strategic principles and planned activities, the US government has established
mechanisms for monitoring NASA's progress. Specialized review bodies, appointed by government authorities, conduct
periodic assessments to evaluate the quality and outcomes of technology transfer initiatives targeted at the private
industrial sector. These reviews include recommendations aimed at expediting and expanding the impact of technology
transfer (Busch, 1996). Such oversight was particularly significant during the initial decade of NASA's existence when
the agency's overall investments reached their peak levels. Within these investments, considerable emphasis was placed
on research and development efforts, leading to notable advancements in crucial technological domains, including
integrated circuits, robotics, radiation utilization and shielding, computational software, telecommunications, and more.
After several years, in the aftermath of the Space Shuttle disasters that cast uncertainty over NASA's exploration
programs, there emerged a strong inclination to bolster initiatives for the commercialization of space activities and,
consequently, facilitate technology transfer. This led to the establishment of the Chief Technology Office in 2010, which
assumed a prominent position in NASA's organizational structure. This office was entrusted with the crucial task of
organizing and overseeing the agency's technology commercialization programs.
In terms of the measures adopted by the US government to support the advancement of technology transfer activities,
particularly the establishment of new ventures, the enactment of the Bayh-Dole Act in 1980 played a pivotal role. This
legislation empowered small businesses and universities to acquire complete or partial ownership of innovations
developed with the use of public funding (Mowery and Sampat, 2004). The implications of this provision extended to
NASA's technology transfer processes as well. As elaborated in the subsequent paragraph, industrial firms operating
outside the realm of space industry, yet expressing interest in adopting NASA's technological breakthroughs, receive
comprehensive assistance encompassing patent support and financial aid. In this regard, NASA has maintained a
longstanding collaboration with the merchant bank Ocean Tomo Federal Services LLC, which commenced in 2008. The


objective of this collaboration is to maximize the economic value of licensed patents for the benefit of the requesting
enterprises (Matthews, 2009).

**3.1 Technology transfer structures**
The significance of technology transfer activity is manifested in its position within the hierarchical structure of the agency.
The transfer programs fall within the jurisdiction of the central entity known as the "Office of Technology and Strategy,"
which directly reports to the Deputy Associate Administrator for Business Operations. The execution of operational
responsibilities is allocated to 10 units situated within NASA, predominantly research centers situated in different states
throughout the nation, each hosting specialized offices (refer to Table 1 below for further details).

| Center Name | Expertise and Activities | Location |
|---|---|---|
| AMES Research Center | Astronomical studies and space technologies (planets and asteroids), supercomputers, robotic missions, automatic control systems for space flights | Silicon Valley, California |
| Armstrong Research Center | Aeronautical flight technologies | Ken County, California |
| Glenn Research Center | Astrophysics research | Cleveland, Ohio |
| Goddard Space Flight Center | Suborbital space flights | Greenbelt, Pennsylvania |
| Jet Propulsion Laboratory | Spacecraft propulsion. JPL is managed by the California Institute of Technology | Pasadena, California |
| Johnson Space Center | Mission preparation and control | Houston, Texas |
| Kennedy Space Center | Spacecraft launches | Cape Canaveral, Florida |
| Langley Research Center | Aeronautical and space studies | Hampton, Virginia |
| Marshall Space Flight Center | Propulsion and communication networks | Madison County, Alabama |
| Stennis Space Center | Propulsion mechanism testing | Hancock County, Mississippi |

**Table 1 - NASA Centers with Technology Transfer Offices**

The specialized transfer offices primarily concentrate on Small Business Innovation Research (SBIR) initiatives, which
aim to facilitate the transfer of NASA technologies to small businesses, thereby promoting the creation of spinoff
enterprises. In the case of Space Transfer Innovation Research (STIR) projects, these offices facilitate the transfer of
technologies to established companies operating in non-space sectors (Gaster, 2017). These projects demand a
significantly higher level of commitment from the transfer offices and often involve engaging in coaching activities. This
entails defining the profile of potential new businesses, which necessitates market evaluations, assessment of financial
resources, and evaluation of the organizational and entrepreneurial capabilities of individuals aspiring to establish new
industrial ventures.
There are indications that NASA's shift towards an ecosystem approach is moving away from a "market creation"
approach and resembling more of a "fixing market failure" approach (Mazzucato, 2015). Upon initial examination, it
appears that NASA has transitioned from being a dominant driver of innovation and development with active mission-
oriented policies (Foray et al., 2012) to adopting diffusion-based policies (Chiang, 1991). In this new approach, NASA's
role is to support the establishment of favorable conditions for markets to emerge, taking on a standard market failure
approach.

**3.2 Partnership as the Paradigm of Technology Transfer in the Third Age of NASA**
Following the Space Shuttle Columbia disaster in 2001, NASA underwent a significant shift in its focus from manned
astrophysical exploration to unmanned missions. This transition led to a notable emphasis on robotic technology and
advancements in radiation-related technologies, which are crucial for astrophysical observations conducted using
telescopes. These technological innovations not only benefited astrophysical research but also had far-reaching
implications for Earth observations. Examples include advancements in spectroscopic investigations, the use of laser
beams, and improvements in terrestrial image resolution.
Due to budget constraints compared to previous phases of development, NASA's technology transfer initiatives often take
place within the framework of partnerships. These partnerships involve collaborations with private or public organizations
within the United States. An illustrative instance is NASA's collaboration with the government of California in a program
aimed at monitoring methane emissions in specific areas of the state's territory. Another example is the collaborative
agreement between NASA and Google, signed on July 1, 2022, which seeks to leverage NASA's satellite data to generate
real-time maps of air pollution levels across various regions of the United States.
NASA's collaboration with other space agencies has been characterized by the partnership paradigm. While these
agreements primarily serve scientific purposes, they often encompass political objectives and address common practical
issues (Lambright and Schaefer, 2004). Several examples highlight this trend. Even during the Space Race era, NASA
forged a strong collaboration with the Japanese space agency (then known as NASDA) to support Japan's economic
recovery as a former defeated nation and subsequent ally of the United States. This collaboration, which involved the
transfer of knowledge with military implications, was established despite Japan's commitment to disarmament outlined
in the 1951 San Francisco Peace Treaty.
NASA's collaboration with the European Space Agency (ESA) has resulted in various astrophysical missions, including
the ongoing "Solar Orbiter" mission aimed at studying the Sun's corona and the origin of magnetic storms. Collaborative
missions focused on Earth observations have also been undertaken, with NASA partnering with agencies from different
countries. For instance, the "Surface Water and Ocean" (SWOT) mission (CNES, 2023), resulting from collaboration
between NASA and the French space agency CNES, aims to accurately measure Earth's surface water topography,
including ocean extent and depth. SWOT, which began its preparatory studies in 2004, seeks to enhance understanding
of climate change's impact on global surface water systems and coastal configurations.
Furthermore, the International Space Station (ISS) stands as a significant achievement resulting from the collaboration of
NASA, the Russian space agency Roscosmos, ESA, and JAXA. These examples of international cooperation by NASA
illustrate the agency's diverse collaborative activities and the broad spectrum of knowledge and experience gained through
these exchanges. It is worth noting that NASA has maintained a strong cooperation with the U.S. Department of Defense
(DoD) as well, which had a budget equivalent to NASA's in 2022. Additionally, NASA is actively collaborating with the
DoD in the establishment of the U.S. Space Force, as announced by President Trump in 2022.
To underscore its engagement with the ESA market, NASA has recently restructured its technology transfer and industrial
property management by establishing the Directorate for Technology Commercialization. This new structure adopts the
partnership paradigm with other organizations, following NASA's model, and actively seeks spin-in opportunities through
dialogue with non-space companies.

**4 ESA's Approach to Technology Transfer**
ESA (European Space Agency) has a dedicated focus on technology transfer and commercialization through its
Technology Transfer and Business Incubation Office (TTPO). The TTPO's mission is to facilitate the transfer of space
technologies and expertise from the space sector to non-space industries, promoting innovation, economic growth, and
societal benefits.
ESA holds a significant portfolio of patents (552 by the end of 2022) resulting from its space research and development
activities. These patents are made available for licensing to interested companies and organizations outside the space
sector.
ESA operates a network of Business Incubation Centers across Europe. These centers provide support, mentoring, and
resources to startups and entrepreneurs aiming to develop and commercialize space technologies in non-space sectors.
ESA supports the development and demonstration of technology transfer projects through its Technology Transfer
Demonstrator Program. Demonstrators showcase the feasibility and market potential of specific technologies, attracting
interest from potential users and investors.
ESA's Business Applications Programs provide funding opportunities for companies and entrepreneurs to develop and
implement innovative applications of space technologies in various sectors. As an example, ESA is willing to provide

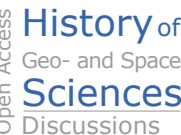

financial support (through the granting of grants up to 50,000 euros) to proponents of space spin-offs. Another initiative
of the agency concerns the Spark Fund, which was established to accelerate the transfer of space technologies to non-
space companies. It serves as support for streamlining the innovation roadmap towards the market. This tool is made
available to the network of technology brokers, who can also benefit from the collaboration of an Investors Network
activated by the ESA Directorate for the commercialization of space technologies (Kinge and Russo, 2000).
ESA facilitates connections between technology seekers and providers through its technology brokerage services. This
involves identifying specific technology needs in non-space industries and connecting them with relevant space-derived
solutions and expertise.
ESA plays a "catalyst" role in the development of new commercial or institutionally-governed operators for specific space
applications domains, which leads to the opening of new markets. An example of this is the development of the SPOT
Earth Observation satellites by the Centre National d'Etudes Spatiales (CNES), which resulted in the creation of the first
commercial operator and dealer for space imagery, SPOT Image. ESA has also contributed to the creation of sector-wide
European entities in areas such as space telecommunications, remote sensing, and space transportation.
While ESA is primarily an R&D entity focused on space science and technology development, it is not its standard
procedure to manage the full operation of a system in the long term. Instead, ESA seeks independent operators to transfer
responsibilities for operations and user engagement, although it may provide support and necessary facilities if needed.
ESA's role is crucial in initiating and developing space applications, demonstrating the technology and outlining its
financial aspects. Once successful, ESA either selects an operator or establishes one. The agency maintains a strong
relationship with the operator, assisting in system integration and providing expertise. The developer agency continues to
act as the procurement source for new generations of space systems.
Once an operator is selected, there is a direct transfer of responsibilities from the development agency to the operator.
The operator is responsible for operating the system, ensuring product and service availability, engaging with users,
identifying requirements, procuring new systems, and ensuring cash-flow. This process leads to the establishment of a
sustainable service. In some cases, the original developer agency still plays a role as a procurement source.
The process of technology transfer and commercialization in ESA's history has occurred in meteorology,
telecommunications, and launchers, with each instance having unique details. Meteorological services and
telecommunications have been maintained as public entities for public goods, while the latter stimulated enough market
growth to be privatized. These entities were initially established as public International Governmental Organizations
(IGOs) in the 1970s and evolved over time.

**4.1 A "network-based" approach to technology transfer**

The technology transfer programs of ESA rely on three interconnected networks operating across Europe, which
distinguishes ESA from NASA. The first network consists of brokers and expert technologists present in 30 European
Union countries. They facilitate the transfer and development of space technologies, including collaborations with
companies in other industrial sectors. This network has fostered spin-in technology transfer projects, particularly by
supporting the growth of small businesses operating within the supply chain of major contractors for exploratory missions.
The second network comprises Business Incubation Centers (BIC), with 90 units across EU countries by the end of 2022
and an annual growth rate expected to be no less than 40 units. BICs primarily focus on generating spin-offs in the satellite
services sector, such as navigation, telecommunications, environmental surveys, and territorial monitoring. Aspiring
entrepreneurs receive a two-year support period and a €50,000 grant for startup expenses. Local authorities often provide
favorable services and financial resources to support the development of BICs, driven by the aim of creating new
employment opportunities.
The NEREUS network, founded in 2007 as an associative structure of European regions, has played a significant role in
facilitating the transfer of space technologies. It connects regions from European Union member states and utilizes
observations from Copernicus SENTINEL satellites for various land surveys and monitoring activities, including water
regimes, agriculture, land subsidence, transportation, forests, and biodiversity. The EU supports the NEREUS network
based on the belief that the development of satellite services can promote economic growth and cohesion among
populations in different territorial areas of the continent. GMES interventions by the NEREUS network have involved 26
regions in 12 EU member countries, along with collaborative relationships with 36 regions from non-European countries.

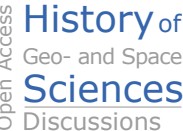

The network has published a collection of 99 case histories that provide analytical information on the results and methods
adopted using Copernicus satellite observations.

**4.2 ESA and local authorities**
ESA maintains a close relationship with local authorities, including regional and municipal governments, within its
member states. This collaboration is crucial for leveraging local resources, expertise, and infrastructure to support ESA's
space programs and initiatives.

ESA recognizes the importance of engaging with local authorities to foster regional development, stimulate innovation,
and create opportunities for economic growth. ESA collaborates with local authorities in various ways:
1. Infrastructure: ESA often relies on existing infrastructure and facilities provided by local authorities for its space-
related activities. This includes the use of launch sites, testing facilities, and research centers located within the regions.
The French Guiana Space Center is a prime example of regional spaceport development due to its strategic location near
the equator, which offers several advantages for launching satellites. The development of the French Guiana Space Center
involved close collaboration between ESA, the French government, and local authorities. ESA has worked in partnership
with the Centre National d'Etudes Spatiales (CNES), the French space agency, to establish and develop the spaceport
infrastructure.
2. Funding and Support: ESA provides financial support to regional projects through its funding programs. Local
authorities can access ESA grants and resources to develop and implement space-related projects and initiatives within
their regions. This funding helps to stimulate local economies, create jobs, and drive innovation. One such example is the
collaboration between ESA and the Regional Innovation and Entrepreneurship Funds (FIRE) in Catalonia, Spain to
support innovation in space-related technologies, applications, and services.
3. Knowledge Exchange: ESA engages in knowledge-sharing activities with local authorities to promote the transfer of
space-related expertise and technology. This includes organizing workshops, seminars, and training programs for local
stakeholders to enhance their understanding of space applications and foster collaboration between ESA and regional
entities. An example of an educational and outreach program conducted by ESA in collaboration with local authorities is
the "ESA School Space Education" initiative, in implementing the "Astro Pi" project. Astro Pi involves sending Raspberry
Pi computers equipped with special sensors and cameras to the International Space Station (ISS).
4. Policy and Regulation: ESA collaborates with local authorities in shaping policies and regulations related to space
activities. This involvement ensures that local regulations align with ESA's objectives and facilitate the growth of space-
related industries within the regions. Local authorities play a vital role in creating an enabling environment for space
innovation and entrepreneurship. One practical example of collaboration between ESA and local authorities for policy
and regulation is the partnership between ESA and the European Union (EU) in the development and implementation of
the Copernicus program. In this collaboration, ESA works closely with local authorities and regulatory bodies within EU
member states to ensure the effective implementation and utilization of Copernicus data and services at the regional and
national levels. For instance, local authorities responsible for environmental management, urban planning, agriculture,
and disaster response can access and utilize Copernicus data and services to monitor land use, assess environmental
changes, manage resources, and enhance preparedness for natural disasters. The data provided by Copernicus satellites
can contribute to evidence-based policymaking and facilitate the development of sustainable strategies at the local level.
5. Socio-economic Impact: ESA works closely with local authorities to assess and promote the socio-economic impact of
space-related activities within their regions. This includes studying the economic benefits, job creation potential, and
societal advancements resulting from space programs. ESA and local authorities collaborate to showcase these impacts
and attract further investments in the space sector. One notable case study is the collaboration between ESA and the
Regional Council of Occitanie in France for the development of precision agriculture applications. Occitanie is an
agricultural region facing challenges related to optimizing crop production, reducing environmental impact, and
improving resource efficiency.
In this collaboration, ESA and the Regional Council of Occitanie worked together to leverage satellite data and
technologies to develop innovative solutions for precision agriculture. Satellite imagery, combined with ground-based
sensors and data analytics, provides farmers and local authorities with real-time information on crop health, soil moisture,

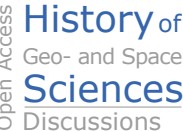

and environmental conditions. The partnership resulted in increased productivity, resource efficiency, and environmental
sustainability in local farming practices. It also showcased the potential for broader adoption of satellite-based
applications in other agricultural regions, leading to socio-economic benefits beyond the initial collaboration.

## 5 Summary of Approaches Adopted

NASA and ESA have collaborated extensively in astrophysics missions and as partners in the International Space Station
(ISS), but they have distinct approaches to technology transfer.
ESA is an association of 15 member states, each represented by their national agencies, while NASA is the sole space
agency of the US federal government. ESA's strategies and commitments are based on objectives shared by its member
states, while collaborative projects involving the military are managed by national space agencies. In contrast, NASA has
collaborative relationships and knowledge exchange with research facilities in the US defence sector.
NASA's technology transfer programs are driven by market logic, aiming to find commercial applications for scientific
knowledge and space technologies. Their focus is on economic development through the creation of tradable products
and services. ESA, on the other hand, prioritizes the ability of space technologies to meet the economic and social needs
of its member states, leading to investments in Earth Observation programs. Upon initial examination, it seems that NASA
has transitioned from being the dominant director of innovation and development, with active mission-oriented policies
to adopting diffusion-based policies. In this new approach, NASA's role is to facilitate the creation of the right conditions
for markets to emerge, adopting a standard market failure approach.
ESA's technology transfer projects involve networks such as NEREUS and BIC, which closely collaborate with regions
and local public administrations in member states. ESA recognizes the importance of public administration structures as
vehicles for technology transfer and diffusion.
NASA often adopts a partnership paradigm, collaborating with other space agencies, large companies, universities, and
individual states within the USA to gain new knowledge and cost savings. ESA also engages in partnerships with space
agencies but has also employed collective partnership systems. ESA encourages potential partners to associate themselves
before initiating cooperative projects, as seen in the creation of the NEREUS network and collaborations with various
associations.
In summary, while NASA's technology transfer is driven by market orientation, ESA prioritizes meeting the needs of its
member states. ESA also emphasizes collaboration with local authorities and adopts collective partnership approaches for
technology transfer. These differences reflect their distinct governance structures and strategic priorities.

## 6 Summary

ESA and NASA have distinct technology transfer approaches and models, although there are also areas of overlap and
similarities. Key differences may be traced back to the following:
1. Organizational Structure: ESA has a dedicated unit called the Technology Transfer and Business Incubation Office
(TTPO) responsible for managing technology transfer activities. NASA, on the other hand, has the Office of the Chief
Technologist (OCT) and the Office of the Chief Technologist - Partnerships (OCT-P) overseeing technology transfer
efforts. The organizational structure and focus of these units may influence the strategies and priorities in technology
transfer.
2. Funding Mechanisms: ESA operates its Technology Transfer Program (TTP), which provides financial support to
promote the transfer of space technologies to non-space sectors. This includes funding for research and development
projects, business incubation centers, and technology demonstrations. NASA, on the other hand, primarily relies on its
internal budget for technology transfer initiatives, with a focus on partnerships and collaborations with industry and other
organizations.
3. Approach to Intellectual Property: ESA tends to adopt a more open approach to intellectual property, often making its
patents available for licensing to interested parties outside the space sector. NASA, on the other hand, has a more diverse

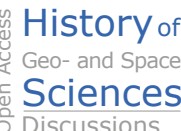

approach, with some technologies being patented and licensed, while others are made available as public domain or subject to specific restrictions.

4. International Collaborations: ESA, as a multinational organization, places emphasis on international collaborations and partnerships with other space agencies and countries. This includes technology transfer initiatives conducted in cooperation with its member states and international partners. NASA, while also engaging in international collaborations, primarily focuses on partnerships within the United States, including collaborations with industry, academia, and government agencies.

5. Business Incubation Centers: ESA operates a network of ESA Business Incubation Centers (ESA BICs) across Europe, providing support and resources to startups and entrepreneurs. These centers help foster the development and commercialization of space technologies in non-space sectors. NASA has a more decentralized approach to business incubation, with various centers and programs supporting entrepreneurship and technology commercialization, but without a centralized network like the ESA BICs.

6. Commercialization Focus: NASA's technology transfer efforts often emphasize commercialization and the transfer of technologies to the private sector for market-driven applications. ESA, while also targeting commercialization, has a broader focus on technology transfer for socio-economic impact, including applications in industry, public services, and societal benefit domains.

It's important to note that despite these differences, both ESA and NASA share common goals of maximizing the societal and economic impact of space technologies, fostering innovation and entrepreneurship, and driving advancements in various sectors through technology transfer. Both agencies recognize the importance of leveraging space technologies for terrestrial applications and promoting collaborations with external entities to achieve these objectives.

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
