# Peer review of "Comparing the evolution of ESA versus NASA technology transfer approach: market and public demand drivers"

_History of Geo- and Space Sciences, 2023_

## Referee Comment (RC2)

**Review of paper**
***"Comparing the evolution of ESA versus NASA technology transfer***
***approach: market and public demand drivers"***

**General comment:**

In reading the paper, one may question different interpretations of the terminology of "technology transfer":
- Transfer from the public domain (i.e. space agencies) to the commercial domain (and for social benefits) (lines 44-47, lines 52-56)
- Transfer from the space area to the non-space area
- Transfer from expert nations to international partners (lines 37-38-39) or developing countries (lines 61-62)
- Transfer from European / national domain to the local / level of the regions
- TBC: transfer from the scientific domain to the application domain ? (lines 74-79)
- TBC: Transfer from the military domain to the civil domain ? (line 80-85)

This is up to the authors, but it could be useful to explain, as above, what are the different types of "technology transfer" that are addressed by this paper. And potentially re-structure the text accordingly.

**Technology transfer at ESA**

As being part of ESA, the following comments focus on the aspects related to technology transfer at ESA only (not at NASA).

The paper only partially describes the technology transfer activities at ESA.

There is a Business incubation service (mentioned at lines 316-321 and 430-434) which enables access to a pipeline of novel commercial enterprises nurtured within the ESA BIC network (https://commercialisation.esa.int/esa-business-incubation-centres/). Up to now there have been 29 BICs with 200 start-ups supported annually and more than 1600 start-ups selected overall.

The ESA InCubed (https://incubed.esa.int/) initiative focuses on developing innovative and commercially viable products and services that generate or exploit the value of Earth observation imagery and datasets. Its fund size is about 182 M€.

The ESA Business Applications and Space solutions (https://business.esa.int/) enable satellite data and technology to transform businesses on Earth. 250M€ have been invested in launching innovative services in over 1200 businesses. This programme could be better covered in lines 278-279.

The ESA General Support Technology Programme is an ESA programme aimed at the development of new and innovative space technologies:

The ESA Phi-Labs network facilitates integration of innovative commercial solutions via the PhilabNet into ESA missions ([https://commercialisation.esa.int/phi-labnet/](https://commercialisation.esa.int/phi-labnet/)). Up to now there have been 12 Phi-labs targets and there 4 currently in development.

The technology brokers ([https://commercialisation.esa.int/technology-broker/](https://commercialisation.esa.int/technology-broker/)) identify technologies that have spin-off potential and support their commercialisation through ESA Spark Funding (the latter is mentioned in lines 280-284). Up to now there have been 9 technology brokers and more than 400 technology transfers.

The ScaleUp Invest initiative ([https://commercialisation.esa.int/2022/10/scaleup-programme-invest-element-the-scaleup-marketplace/](https://commercialisation.esa.int/2022/10/scaleup-programme-invest-element-the-scaleup-marketplace/)) creates a marketplace for suppliers and customers to secure space services at competitive prices. It currently supports companies with the possibility to access ~ 25M€ in funding for IOD/IOV deals.

The ESA Investor network ([https://commercialisation.esa.int/investor-network/](https://commercialisation.esa.int/investor-network/)) builds effective partnerships with capital allocators, key institutions, accelerators and investors and comprises more than 250 investors.

The ESA ambassador network ([https://business.esa.int/ambassador-platforms](https://business.esa.int/ambassador-platforms)) extends the reach of ESA and space-companies into non-space sectors through dedicated in-country support. It has currently 13 ambassadors in 7 Member States of ESA.

The final stages of commercialisation also include initiatives like EO data buy (for science through Earthnet and for operations through Copernicus) and the ESA Newspace approach with the funding of the Scouts or AWS missions.

A brief description of the above initiatives in chapter 4 could give a more complete picture of the technology transfer & commercialisation programmes at ESA.

**Additional comments:**

[Regarding one comment and its reply posted about the paper]
On the statement: "*In the technical-scientific and economic literature, "upstream" space activities typically concern Outer Space launch and exploration systems while down-stream activities typically concern Earth Observations and the services that are generated by them such as, for example, satellite services (Navigation, telecommunications, etc.)."*

At ESA we rather consider "Upstream" the activities linked to the development of space mission / satellites, whereas downstream is more related to activities linked to the exploitation of the space systems and their data.

It seems the terminology "Earth observation" is used for system orbiting around the earth (i.e. incl. navigation, telecommunication), whereas we consider at ESA that the terminology "Earth

Observation" is related to the space systems observing the earth, i.e. remote sensing. In that respect the sentences lines 121-122-123 are not fully clear.

Lines 254-255: the cooperation on the ISS also includes Canada / CSA.

Lines 257-258: the equivalence of budget between NASA and the US DoD is most probably related to the space activities of the DoD.

Lines 260-261: please clarify the involvement of NASA in the "ESA market". Is "ESA" wrongly inserted here ? Is it rather meant "commercial market" ?

Lines 275-277: The authors mention ESA Technology Transfer Demonstration Programme. To be clarified whether this refers to the Technology Transfer Programme which was created to facilitate technology transfer from space technologies to terrestrial applications and to help with the commercialisation of such applications.
(https://www.esa.int/Enabling_Support/Space_Engineering_Technology/Technology_Business_Opportunities/Technology_Transfer_Programme)

Lines 278-280: the authors are invited to consult: https://business.esa.int/
There are different initiatives, from non-equity funding (up to 2M€) to tailored project management support to businesses concerning business applications programmes. The level of funding depends on the type of activities (i.e. feasibility studies in open competition or direct negotiation, kick-start studies or demonstration projects) and range from 50K to 400K€

Lines 293-294-295: this is true, but to some extent. The ESA convention does not prevent ESA to perform operations of space systems. It is true that the meteorological scheme whereby ESA develops satellites and transfer them to EUMETSAT for their operations is a good illustration of what is written. However, ESA is also responsible for the operations of large systems with many satellites, like in the case of Copernicus, with funding delegated from the EU. The operations of the Sentinel satellites are shared between ESA and EUMETSAT.
These are just remarks to the authors, no need to amend the text.

Line 325: it is suggested to add "ESA", i.e.: "The EU and ESA support the NEREUS..."
Indeed the described activity have been made under an ESA contract, with funding from the EU.
This is an excellent example for indeed promoting the economic growth at the level of the regions using Copernicus Sentinel data. The use of Sentinel data at regional and local levels also allows the public authorities to use space-derived technology to combat the effect of climate change, sustainable agriculture, water management, forestry monitoring etc. This also helps the economy with the creation of related value-added services. This is actually very well covered under point 4. of chapter 4.2.

Lines 368-374: For the socio-economic impact, and showing the benefits brought by the usage of Sentinels data to society, environment and economy, please be aware of the activity ESA has been organizing with EARSC on Copernicus: https://earsc.org/sebs/

Line 385: ESA has currently 22 Member States.
Note that not all MSs have a space agency, so suggest to add: ".... each represented by their national agencies or relevant ministries."

Line 392: not only related to Earth Observation but space application systems more generally. It is true that ESA being an inter-governmental organization, it performs the activities primarily for the interests of its Member States (as well as for the EU for joint programmes like Copernicus).
However ESA also launches on its own specific initiatives to foster the commercialization linked to space, in both the upstream (space segment, satellites) and downstream (data exploitation), that in fine serve also its Member States.

---

## Author Comment (AC4)

**Review of paper "Comparing the evolution of ESA versus NASA technology transfer approach: market and public demand drivers"**

The authors wish to express sincere appreciation and gratitude to the referee for the invaluable and insightful comments. The feedback is highly appreciated, not only for its evident usefulness but also because it emanates from a direct and experienced involvement in the field. We hope that our replies to the thoughtful insights may contribute significantly to the enhancement of the quality and depth of our work.

| REVIEWER'S COMMENTS | OUR REPLY |
|---|---|
| In reading the paper, one may question different interpretations of the terminology of "technology transfer":
 - Transfer from the public domain (i.e. space agencies) to the commercial domain (and for social benefits) (lines 44-47, lines 52-56)
 - Transfer from the space area to the non-space area
 - Transfer from expert na tions to international partners (lines 37-38-39) or developing countries (lines 61-62)
 - Transfer from European / national domain to the local / level of the regions
 - TBC: transfer from the scientific domain to the application domain ? (lines 74-79)
 - TBC: Transfer from the military domain to the civil domain ? (line 80-85)
 This is up to the authors, but it could be useful to explain, as above, what are the different types of "technology transfer" that are addressed by this paper. And potentially re-structure the text accordingly. | AGREED UPON THIS POINT.  WE WOULD ADD AT THE END OF PARAGRAPH 2 THE FOLLOWING: " As sketched above, technology transfer can be interpreted in various ways, including the shift from the public domain (e.g., space agencies) to the commercial and social sectors, transition from space to non-space domains, transfer between expert nations and developing countries, movement from European/national to local/regional levels, bridging the gap between scientific and application domains, and transitioning from military to civil domains. In this paper, the concept of technology transfer is specifically addressed with a primary focus on its practical implications for the commercial and social sectors." |
| [Regarding one comment and its reply posted about the paper]
 On the statement: "In the technical-scientific and economic literature, "upstream" space activities typically concern Outer Space launch and explora-on systems while down-stream activities typically concern Earth Observations and the services that are generated by them such as, for example, satellite services (Navigation, telecommunications, etc.)."
 At ESA we rather consider "Upstream" the activities linked to the development of space mission / satellites, whereas downstream is more related to activities linked to the exploitation of the space systems and their data. | AGREED UPON THIS POINT. WE HAVE CLARIFIED THE DISTINCTION IN OUR REPLY POSTED AS FOLLOWS: "In the realm of space endeavors, "upstream" denotes the array of activities intricately connected with the evolution and creation of space missions and satellites. Conversely, "downstream" pertains more to the activities associated with the utilization and operation of space systems, as well as the processing and analysis of the data they generate." |
| It seems the terminology "Earth observation" is used for system orbiting around the earth (i.e. incl. navigation, telecommunica tion), whereas we consider at ESA that the terminology "Earth | AGREED UPON THIS POINT. ON LINE 121, WE WOULD REPLACE "In the last 15 years, there has been a notable acceleration in the widespread adoption of Earth |

| | |
|---|---|
| Observation" is related to the space systems observing the earth, i.e. remote sensing. In that respect the sentences lines 121-122-123 are not fully clear. | Observation activities." WITH In the last 15 years, there has been a notable acceleration in downstream activities". |
| Lines 254-255: the cooperation on the ISS also includes Canada/CSA. | AGREED UPON THIS POINT. WE WOULD ADD "Canada/CSA". |
| Lines 257-258: the equivalence of budget between NASA and the US DoD is most probably related to the space activities of the DoD. | AGREED UPON THIS POINT. WE WOULD ADD "…which had a budget for space activities equivalent to NASA's in 2022". |
| Lines 260-261: please clarify the involvement of NASA in the "ESA market". Is "ESA" wrongly inserted here ? Is it rather meant "commercial market" ? | AGREED UPON THIS POINT. ON LINE 260, WE WOULD REPLACE ""ESA market" with "commercial market". |
| Lines 275-277: The authors men tion ESA Technology Transfer Demonstration Programme. To be clarified whether this refers to the Technology Transfer Programme which was created to facilitate technology transfer from space technologies to terrestrial applications and to help with the commercialisation of such applications. | AGREED UPON THIS POINT.  WE WOULD SPECIFY THAT: "ESA supports the development and demonstration of technology transfer projects through its Technology Transfer Demonstrator Program, an open call for proposals made by the The Technology Transfer Programme Office (TTPO). These projects are aimed at assessing and mitigating technical risks associated with adapting established space technologies for novel terrestrial applications. These initiatives encompass the creation and validation of innovative hardware and software derived from European Space programs, enhancing the viability of transitioning core technologies from space to terrestrial environments. ESA is committed to providing financial support, contributing up to €39,000 for each selected proposal to assist in covering associated costs." |
| The paper only partially describes the technology transfer activities at ESA. A brief description of the above (following in this note) initiatives in chapter 4 could give a more complete picture of the technology transfer & commercialisation programmes at ESA. | |
| There is a Business incubation service (mentioned at lines 316-321 and 430-434) which enables access to a pipeline of novel commercial enterprises nurtured within the ESA BIC network (https://commercialisation.esa.int/esa-business-incubation-centres/). Up to now there have been 29 BICs with 200 start-ups supported annually and more than 1600 start-ups selected overall. The ESA InCubed (https://incubed.esa.int/) initiative focuses on developing innovative and commercially viable products and services that generate or exploit the value of Earth observation imagery and datasets. Its fund size is about 182 M€. | AGREED UPON THIS POINT.  WE WOULD ADD AND SPECIFY THAT: The Business Incubation Service provides a gateway to a diverse array of emerging commercial ventures cultivated within the extensive ESA BIC network. To date, the network boasts 29 Business Incubation Centers (BICs), fostering 200 start-ups annually and supporting a cumulative total of over 1600 start-ups. In tandem, the ESA InCubed initiative takes center stage, dedicated to the advancement of innovative and economically sustainable products and services. The primary focus lies in harnessing the value inherent in Earth observation imagery and datasets. With a substantial fund size of approximately €182 million, InCubed is poised to catalyze the |

| | development of groundbreaking ventures in this dynamic space. |
|---|---|
| The ESA Business Applications and Space solutions (https://business.esa.int/) enable satellite data and technology to transform businesses on Earth. 250M€ have been invested in launching innovative services in over 1200 businesses. This programme could be better covered in lines 278-279. | AGREED UPON THIS POINT. WE WOULD ADD AND SPECIFY THAT: The ESA Business Applications and Space Solutions programme empowers businesses on Earth to undergo transformation through the utilization of satellite data and technology. A substantial investment of €250 million has been dedicated to the launch of pioneering services, benefiting over 1200 businesses. |
| The ESA General Support Technology Programme is an ESA programme aimed at the development of new and innovative space technologies. The ESA Phi-Labs network facilitates integration of innovative commercial solutions via the PhilabNet into ESA missions (https://commercialisation.esa.int/phi-labnet/). Up to now there have been 12 Phi-labs targets and there 4 currently in development. | AGREED UPON THIS POINT. WE WOULD ADD AND SPECIFY THAT: "The ESA General Support Technology Programme is geared towards fostering the creation of novel space technologies. Concurrently, the ESA Phi-Labs network plays a crucial role in seamlessly integrating innovative commercial solutions into ESA missions through the PhilabNet. To date, 12 Phi-Labs targets have been identified, with four currently in the developmental pipeline." |
| The technology brokers (https://commercialisation.esa.int/technology-broker/) identify technologies that have spin-off potential and support their commercialisation through ESA Spark Funding (the latter is mentioned in lines 280-284). Up to now there have been 9 technology brokers and more than 400 technology transfers. | AGREED UPON THIS POINT. WE WOULD ADD AND SPECIFY THAT: "The technology brokers are tasked with pinpointing technologies with spin-off potential and aiding in their commercialization through the utilization of ESA Spark Funding, as outlined in lines 280-284. To date, there are nine technology brokers actively involved, contributing to the successful execution of over 400 technology transfers." |
| The ScaleUp Invest initiative (https://commercialisation.esa.int/2022/10/scaleupprogramme- invest-element-the-scaleup-marketplace/) creates a marketplace for suppliers and customers to secure space services at competitive prices. It currently supports companies with the possibility to access ~ 25M€ in funding for IOD/IOV deals. | AGREED UPON THIS POINT. WE WOULD ADD AND SPECIFY THAT: "The ScaleUp Invest initiative has pioneered a marketplace connecting suppliers and customers, streamlining the acquisition of space services at competitive rates. Currently, it extends support to companies by offering access to around €25 million in funding for "In-Orbit Demonstration," and "In-Orbit Validation"deals." |
| The ESA Investor network (https://commercialisation.esa.int/investor-network/) builds effective partnerships with capital allocators, key institutions, accelerators and investors and comprises more than 250 investors. | AGREED UPON THIS POINT. WE WOULD ADD AND SPECIFY THAT: "The ESA Investor Network cultivates meaningful collaborations with capital allocators, vital institutions, accelerators, and investors, encompassing a robust community of over 250 financial backers." |
| The ESA ambassador network (https://business.esa.int/ambassador-platforms) extends the reach of ESA and space-companies into non-space sectors through dedicated in-country support. It has currently 13 ambassadors in 7 Member States of ESA. | AGREED UPON THIS POINT. WE WOULD ADD AND SPECIFY THAT: "The ESA Ambassador Network plays a pivotal role in extending the influence of ESA and space-related companies into non-space sectors, offering dedicated in-country support. Currently, the network boasts 13 ambassadors spread across 7 Member States of ESA." |
| The final stages of commercialisation also include initiatives like EO data buy (for science through Earthnet and for operations through Copernicus) | AGREED UPON THIS POINT. WE WOULD ADD AND SPECIFY THAT: "The concluding phases of commercialization encompass endeavors such as Earth Observation (EO) data acquisition, both for |

| | |
|---|---|
| and the ESA Newspace approach with the funding of the Scouts or AWS missions. | scientific purposes through Earthnet and operational needs via Copernicus. Additionally, the ESA NewSpace approach involves funding missions like Scouts or AWS to further advance space exploration." |
| Lines 278-280: the authors are invited to consult: https://business.esa.int/
There are different initiatives, from non-equity funding (up to 2M€) to tailored project management support to businesses concerning business applications programmes. The level of funding depends on the type of activities (i.e. feasibility studies in open competition or direct negotiation, kick-start studies or demonstration projects) and range from 50K to 400K€ | AGREED UPON THIS POINT.  WE WOULD ADD AND SPECIFY THAT: "Diverse initiatives are in place, offering non-equity funding of up to €2 million and providing tailored project management support for businesses engaged in business applications programs. The allocated funding varies based on the type of activities, such as feasibility studies conducted through open competition or direct negotiation, kick-start studies, or demonstration projects, with financial assistance ranging from €50,000 to €400,000." |
| Lines 368-374: For the socio-economic impact, and showing the benefits brought by the usage of Sentinels data to society, environment and economy, please be aware of the activity ESA has been organizing with EARSC on Copernicus: https://earsc.org/sebs/ | AGREED UPON THIS POINT.  WE WOULD ADD AND SPECIFY THAT: "The European Association of Remote Sensing Companies (EARSC) has beeb conducting a comprehensive project, under the assignment from the European Space Agency (ESA) and funded by the European Commission as part of the Copernicus Programme. The project has focuses on assessing the benefits of using Copernicus Sentinel data across various market applications. Through 20 case studies, spanning sectors like precision agriculture, winter navigation, and flood management, the project tracks the impact of satellite data throughout the value chain. The assessment reveals significant value generated through satellite-based Earth Observations, showcasing enhanced productivity, efficient and environmentally friendly operations, economic gains, and improved quality of life. These findings serve as valuable evidence for policy makers, space agencies, public agencies, and companies, demonstrating the tangible benefits of investing in the Copernicus program and leveraging EO data and services." |
| Line 385: ESA has currently 22 Member States. Note that not all MSs have a space agency, so suggest to add: ".... each represented by their na tional agencies or relevant ministries." | AGREED UPON THIS POINT.  WE WOULD ADD. |
| Line 392: not only related to Earth Observation but space application systems more generally. It is true that ESA being an inter-governmental organization, it performs the activities primarily for the interests of its Member States (as well as for the EU for joint programmes like Copernicus). | AGREED UPON THIS POINT.  WE WOULD ADD. |

| | |
|---|---|
| However ESA also launches on its own specific initiatives to foster the commercialization linked to space, in both the upstream (space segment, satellites) and downstream (data exploitation), that in fine serve also its Member States. | |